# The Vanilloid (Capsaicin) Receptor TRPV1 in Blood Pressure Regulation: A Novel Therapeutic Target in Hypertension?

**DOI:** 10.3390/ijms24108769

**Published:** 2023-05-15

**Authors:** Arpad Szallasi

**Affiliations:** Department of Pathology and Experimental Cancer Research, Semmelweis University, 1085 Budapest, Hungary; szallasi.arpad@semmelweis.hu

**Keywords:** vanilloid (capsaicin) receptor TRPV1, blood pressure regulation, hypertension

## Abstract

Today’s sedentary lifestyle with excess food and little exercise increases the number of people with hypertension, a major risk factor for stroke. New knowledge of treatments in this field is of utmost importance. In animal experiments, the activation by capsaicin of TRPV1-expressing sensory afferents evokes a drop in blood pressure by triggering the Bezold–Jarisch reflex. In hypertensive rats, capsaicin reduces blood pressure. Conversely, genetic ablation of the TRPV1 receptor results in elevated nocturnal (but not diurnal) blood pressure. These observations imply a therapeutic potential for TRPV1 activation in hypertensive patients. Indeed, in a major epidemiological study involving 9273 volunteers, dietary capsaicin was found to lower the risk for hypertension. New research indicates that the mechanism of action of capsaicin on blood pressure regulation is far more complex than previously thought. In addition to the well-recognized role of capsaicin-sensitive afferents in blood pressure regulation, TRPV1 seems to be expressed both in endothelial cells and vascular smooth muscle. This review aims to evaluate the therapeutic potential of TRPV1-targeting drugs in hypertensive patients.

## 1. Introduction

Today’s sedentary lifestyle with too much food and too little exercise increases the number of people with abnormally high blood pressure [1]. Nearly half of all adult Americans already have pre-hypertension or hypertension, defined as systolic blood pressure over 130 mmHg, diastolic blood pressure greater than 80 mmHg, or taking medication for hypertension [2]. Even more worrisome is the prevalence of pre-hypertension and hypertension among adolescents: 15.7% and 3.2%, respectively [3]. High blood pressure is a known risk factor for a number of deadly diseases, ranging from stroke through kidney failure to breast cancer. It is generally agreed that lifestyle changes may decrease or even normalize blood pressure in many patients, reducing or negating the need for medications. Nutraceuticals (pharmacologically active dietary supplements) are gaining attention as safe (presumably free of side-effects) alternatives to antihypertensive drugs [4]. The European Society of Hypertension has already issued a position document on nutraceuticals and blood pressure control [5].

Capsaicin is best recognized as the active ingredient in hot chili peppers. Capsaicin can be consumed as a nutraceutical, and hot pepper be eaten as functional food. An estimated quarter of the world’s population is consuming hot pepper on a daily basis. Large epidemiological studies imply health benefits for chili lovers, including lower risk for obesity and cardiovascular disease [6,7,8,9]. Despite intensive research, the molecular mechanisms by which capsaicin may exert these health benefits are yet to be determined [10,11].

Capsaicin is unique among natural irritants in that the initial excitation by capsaicin of sensitive neurons is followed by a lasting refractory state, traditionally termed “desensitization”, in which the previously excited neurons are refractory even to stimuli unrelated to capsaicin [12,13,14]. The pepper plant uses capsaicin as a chemical deterrent to keep away herbivores. Therefore, the human fondness of hot spicy food is puzzling. Clearly, chili pepper preference is an acquired taste with deep cultural roots [15].

Capsaicin evokes a “hot”, burning sensation in the human mouth [12,13]. This is not by accident as the receptor for capsaicin (now known as Transient Receptor Potential, Vanilloid-1, or briefly TRPV1) is also activated by heat [16]. The discovery of molecular heat sensation by TRPV1 has earned a shared Nobel prize in Physiology and Medicine for David Julius [17]. Of note, the pivotal role of TRPV1 in noxious heat sensation was recently confirmed in two individuals carrying a non-functional channel due to missense mutations in the *TRPV1* gene [18].

Based on a meta-analysis of seven clinical trials involving 363 study subjects, fermented red pepper paste was recommended as a food additive to lower abnormally high blood pressure [19]. In a major epidemiological study involving 9273 volunteers in China, dietary capsaicin was found to lower the risk of developing hypertension [20]. The literature on capsaicin and blood pressure is vast (973 papers in PubMed), but the published data are often conflicting. For example, another meta-analysis of clinical trials failed to identify any effect of capsaicin (hot pepper) consumption on blood pressure [21].

This review aims to (1) dissect the role of TRPV1 in physiological blood pressure regulation, and (2) evaluate the therapeutic potential of TRPV1-targeting drugs in hypertensive patients.

## 2. Animal Studies: The Biphasic Effect of Capsaicin on Blood Pressure

The pioneering Hungarian pharmacologist, Endre Hőgyes, is credited with the first efforts to determine the effects of capsaicin on the heart and circulation [22,23]. In his experiments, he used an oily *Capsicum annuum* extract obtained by fellow chemists, Fleischer and Ember-Bogdán. In dogs, he administered this “pure pepper oil” (that he called “capsicol”) via intragastric catheter and noted the concomitant development of bradycardia and slow breathing rate that lasted for about an hour [22]. At the same time, the rectal temperature of the animal dropped from 39.8 °C to 38.9 °C. This was no doubt the first experimental demonstration of capsaicin-evoked Bezold–Jarisch reflex and hypothermia!

He also tested his capsicol in guinea-pigs injected into the jugular vein: although the treated animals became sleepy and inactive, there was no change in their respiratory movements, heart rhythm or rectal temperature [23]. [Parenthetically, Hőgyes called his experimental animals “tengeri nyúl” (“rabbit of the sea”) which is really guinea-pig and not rabbit, as was later translated erroneously.] From his observations, Hőgyes correctly deduced that capsicol is a selective irritant of sensory nerves that evokes a “spicy, hot feeling” [22].

Of note, Hőgyes also tested this “pepper oil” on himself and his assistant professor: he described a warm feeling in the abdomen, followed by unpleasant reflux, flatulence and mild diarrhea [22]. No change in heart rate or respiration was noted [22].

Intrigued by Hőgyes’ initial observations, more than a half century later a Hungarian physiologist, János Pórszász, set out to systematically explore the cardiovascular effects of pure capsaicin in cats and dogs [24]. In intact cats, capsaicin injected i.v. at a dose of 200 μg/kg produced a temporary (lasting for 5 s) drop in blood pressure (45 mmHg), accompanied by bradycardia and apnea. Thereafter, a lasting increase in blood pressure developed (42 mgHg over baseline). Atropine (that blocks the efferent function of the vagus nerve) prevented the initial depressor phase, but the lasting pressor response remained unchanged following atropine, vagotomy or decapitation. Injected directly into the carotid artery or the brain (cisterna magna), capsaicin only evoked the hypertensive response. Based on these results, Pórszász and co-workers reached two major conclusions: (1) capsaicin initially reduces the blood pressure by triggering the Bezold–Jarisch reflex; and then (2) it elevates the blood pressure by direct vasoconstrictor action [24].

Pórszász also tested i.v. capsaicin on dogs. Dogs were more sensitive to capsaicin than cats: a lower capsaicin dose (50 μg/kg) produced a more profound drop in blood pressure (90 mmHg) [24]. Dogs, however, were missing the second hypertensive phase of capsaicin administration, indicating marked species-related differences in cardiovascular capsaicin actions [24].

In the rat, Makara and colleagues [25] described a biphasic capsaicin action on blood pressure similar to that seen in cats [24]. Animals desensitized to s.c. capsaicin (4, 8 and 16 mg per rat, followed by a dose of 200 mg/kg) showed no difference in blood pressure as compared to solvent controls [25]. In rats desensitized to capsaicin, i.v. capsaicin administration (0.5 to 2 μg/kg) evoked a much-attenuated hypotensive action, with no change in the pressor response. Authors concluded that the target of the depressor response was more sensitive to capsaicin desensitization than the target of the pressor response [25].

In 1972, Toda and associates demonstrated that in vitro capsaicin (0.1 μg/mL) can induce a sustained contraction of the isolated superior mesenteric or coronary artery of the dog which was dependent on the presence of Ca^2+^ in the perfusion medium [26]. In the discussion, the authors noted that “capsaicin causes constriction through a mechanism by which the flux of Ca^2+^ into smooth muscle cells is increased” [26]; anticipating the existence of TRPV1 (a non-selective cation channel with preference for Ca^2+^) in vascular smooth muscle by more than two decades.

By this, all the pieces of the capsaicin puzzle were on the table: two anatomically distinct capsaicin targets (one neuronal and another in smooth muscle) and a molecular mechanism that allows Ca^2+^ influx into the vascular smooth muscle.

However, for the below reasons, it took several decades for the puzzle to be solved:Capsaicin was recognized as a selective activator of a well-defined subset of sensory neurons [27,28,29];The initial excitation by capsaicin of these neurons is followed by a lasting refractory state, traditionally referred to as desensitization [30,31,32];The capsaicin-evoked drop in blood pressure was absent in animals desensitized to capsaicin [25];By contrast, capsaicin desensitization had no effect on capsaicin-induced hypertension [25].

Thus, one may argue that capsaicin causes a drop in blood pressure by activating its receptor on sensory neurons, whereas the hypertensive action is a non-specific capsaicin effect.

## 3. Capsaicin Causes Transient Hypotension by Triggering the Bezold–Jarisch Reflex

Clinically, the Bezold–Jarisch reflex (BJR) is an inhibitory reflex triad defined as bradycardia (slow heart rate), hypotension (abnormally low blood pressure) and hypopnea (slow and/or shallow respiration) [33,34]. Originally described by Albert von Bezold in 1867 [35] following i.v. injection of veratrum alkaloids in experimental animals, it remained an oddity of physiology until rediscovered by Adolf Jarisch in 1937 [36]. Now the BJR is recognized as an important protective mechanism to dilate coronary arteries during acute myocardial infarction [37]. On the contrary, the BJR may be responsible for circulatory collapse, a dreaded complication of spinal anesthesia [38].

Although the BJR was first thought to originate from receptors located in pulmonary vessels (hence the alternative terminology, “pulmonary chemoreflex”) [25], recent evidence implicates cardiac inhibitory mechanoreceptors in the left ventricle as triggers of the reflex pathway [39]. Regardless of the trigger point, the efferent pathway of the BJR is no doubt mediated by unmyelinated C-fibers in the vagal nerve. The capsaicin-evoked BJR was eliminated by both atropine and surgical vagotomy [24]. Importantly, the BJR was also absent in TRPV1-null mice [40].

Capsaicin triggers the full triad of the BJR in mice [41], rats [25], dogs [22,24,42,43] and cats [24] but not in guinea pigs [22,23]. In the latter species, capsaicin (5–20 μg/kg i.v.) provokes only a brief (6–9 s) drop in blood pressure, but no bradycardia [44].

Resiniferatoxin is an ultrapotent capsaicin analogue [45] with a unique spectrum of pharmacological actions [46]. Unlike capsaicin, resiniferatoxin (up to 5 μg/kg i.v.) did not trigger the BJR in the rat [47]; however, it rendered the reflex pathway unresponsive to subsequent capsaicin or phenyldiguanide challenge [47]. The failure of resiniferatoxin to provoke the BJR, which is the main limiting factor in the use of capsaicin [48], implied an important advantage to use resiniferatoxin for therapeutic purposes. Indeed, rats can be fully desensitized by a single s.c. resiniferatoxin injection against both neurogenic inflammation [45] and sciatic nerve ligation-induced thermal hyperalgesia [49] without causing respiratory arrest. For capsaicin, multiple injections with increasing doses are needed to achieve a similar effect [50].

In dogs, intrathecal resiniferatoxin (0.1 to 3.0 μg/kg) does not initiate the BJR [51]. Instead, the mean arterial pressure rose from 79 to 131 mmHg within 5 min following resiniferatoxin administration; dropped to 80 mmHg by 50 min; and remained elevated during the whole 240 min of the monitoring [51]. A concomitant increase in heart rate (from 89 to 139 beats/min) was also noted that paralleled the change in blood pressure [51]. In this study, no attempt was made to interpret the resiniferatoxin-induced changes in hemodynamic parameters.

## 4. Capsaicin Evokes Hypertension by Activating TRPV1 Expressed on Vascular Smooth Muscle

It has been long recognized that capsaicin can induce smooth muscle contractions in vitro in bronchi [52,53,54], intestinal tissue [55,56] and vasculature [57], but it was attributed by most authorities to an indirect, nerve-mediated capsaicin action rather than a direct action on muscle cells. Indeed, antagonists of the receptor for substance P (a neuropeptide released from capsaicin-sensitive afferents upon stimulation) were shown to prevent the smooth muscle contraction induced by capsaicin [58].

In the isolated and working rat heart, capsaicin (1 nM to 1 µM) induced a dose-dependent decrease in coronary blood flow [59,60]. Since this effect was mimicked by endothelin, and blocked by the non-specific endothelin receptor inhibitor, PD142893 (200 nM), the capsaicin action was attributed to endothelin released by capsaicin from sensory cardiac afferents [59,60]. Endothelin is a major vasoconstrictor peptide produced by vascular endothelial cells [61], implicated in the pathomechanism of essential hypertension [62]. There is, however, no good evidence to suggest endothelin synthesis in capsaicin-sensitive neurons. In accord, in the guinea-pig, ileum endothelin does not interfere with capsaicin actions [63].

Preliminary evidence for a direct capsaicin action on vascular smooth muscle was also presented. For example, capsaicin was reported to induce a contractile response in the guinea-pig carotid artery and feline middle cerebral artery which was seen even after chronic capsaicin treatment [64]. Furthermore, capsaicin (0.1 μg/mL) was found to induce a sustained contraction of the isolated superior mesenteric or coronary artery of the dog in a Ca^2+^-dependent fashion [26]. Capsaicin also constricted the isolated canine mesenteric artery with an EC_50_ of 3 µM [65]. Finally, capsaicin (1 to 10 µM) decreased blood flow through the medial meningeal artery [66], implying a role for the vascular capsaicin target in headaches.

In cultured rat aortic smooth muscle cells, capsaicin (up to 300 μM) was found to inhibit ^45^Ca^2+^ uptake [67] in sharp contrast to neurons that accumulate massive amounts of Ca^2+^ in response to capsaicin [68,69]. Since the capsaicin receptor TRPV1 is a non-selective Ca^2+^ channel, the inhibitory capsaicin action on Ca^2+^ uptake in smooth muscle implied a non-TRPV1-mediated capsaicin action. Indeed, cultured rat aortic smooth muscle cells do not express functional TRPV1 [70]. In accord, capsaicin had no contractile effect on dog aorta strips [26]. Combined, these observations imply that functional TRPV1 is not expressed in the aorta.

A series of papers from the laboratory of Attila Tóth, however, indicated that TRPV1 may be expressed in smaller arteries and arterioles [71,72,73,74]. In the isolated perfused rat hind limb, intraarterial capsaicin infusion increased vascular resistance by almost 100 mmHg. In isolated gracilis muscle arterioles, capsaicin evoked a dose-dependent biphasic response: (1) vasodilation at low concentrations (up to 10 nM) and (2) vasoconstriction at high concentrations (0.1 to 1.0 μM). Endothelium removal, or nitric oxide synthase (NOS) inhibition, blocked the dilator response but had no effect on the contractile response. Importantly, both TRPV1-like immunoreactivity and TRPV1 mRNA were demonstrated in the muscularis propria of the blood vessels. The distribution of vascular TRPV1 (skeletal muscle, skin, carotis and aorta) implied a role for TRPV1 in blood pressure regulation [73]. This contrasts with the far more discreet vascular TRPV1 expression by another group [75].

This group used a TRPV1 reporter mouse (TRPV1^PLAP-nlacZ^) to demonstrate TRPV1 expression: no signal was seen in the aorta [75]. By RT-PCR, TRPV1 mRNA was restricted to small- and medium-diameter vessels (up to 100 μΜ in diameter), suggesting a role in thermoregulation but not in blood pressure regulation [75]. In arterioles isolated from the ear, capsaicin (1 to 10 μM) induced Ca^2+^ influx and resultant muscle constriction [75].

Two recent studies using reporter mouse lines TRPV1^PLAP-nlacZ^ and TRPV1-Cre:tdTomato combined with Ca^2+^ imaging provided unequivocal proof for TRPV1 expression in small resistance arteries and terminal arterioles supplying the heart, skeletal muscle and adipose tissue (Figure 1) [76,77]. Capsaicin constricted arterioles in vitro and increased mean arterial pressure in vivo. Pharmacological blockade or genetic disruption of TRPV1 abolished the capsaicin effect on vascular smooth muscle [76,77,78]; however, ablation of the sensory afferents had no effect on the capsaicin-evoked hypertension [76,77]. Overall, these observations are in excellent agreement with previous studies in rats [25] and dogs [24].

To some degree, these differences in TRPV1 protein expression may reflect differences in the sensitivity and specificity of the anti-TRPV1 antibodies used in these studies.

Though it is beyond the scope of this review, it should be mentioned briefly that TRPV1 has also been implicated in the proliferation and migration of vascular smooth muscle cells, linking TRPV1 to atherosclerosis and other vascular diseases [79]. For instance, capsaicin dose-dependently increased cytoplasmic Ca^2+^ uptake and blocked lipid accumulation in vascular smooth muscle cells in wild-type, but not in TRPV1-null mice [80].

## 5. Another Player Emerges: TRPV1 in Vascular Endothelium?

As discussed above, the effects of TRPV1 activation by capsaicin of vagal afferents mediating the BJR and vascular smooth muscle are now well-established: transient hypotension and sustained hypertension, respectively. Less clear is the role of endothelial TRPV1 expression in physiological blood pressure regulation.

In endothelial cells isolated from human cerebrovascular tissue, functional TRPV1 was demonstrated by a combination of immunostaining, RT-PCR and Ca^2+^ imaging [81]. It was speculated that these receptors may mediate some of the endocannabinoid actions on brain microcirculation [81]. The potent and selective cannabinoid receptor agonist, Win 55,212-2, induces relaxation of rat aorta rings under control conditions but not following capsaicin desensitization; this implicates TRPV1 in the vascular response [82].

Calcitonin gene-related peptide (CGRP) is a sensory neuropeptide, stored in and released from the peripheral endings of capsaicin-sensitive neurons upon stimulation [83]. CGRP is an important part of the biochemical cascade, collectively known as neurogenic inflammation [84,85]. By its action on arterioles, CGRP is primarily responsible for the erythema (skin reddening) response to topical capsaicin administration [85]. Interestingly, CGRP is also produced by vascular endothelium where, similar to sensory neurons, its release is modulated by TRPV1 [86].

TRPV1 is expressed in the rat coronary artery where it is thought to mediate Ca^2+^-dependent nitric oxide (NO) release [87]. In the mouse kidney, TRPV1 is predominantly expressed in the narrow resistance vessels where capsaicin evokes endothelium-dependent vasorelaxation [88].

In Sprague-Dawley rats, a triphasic blood pressure response to i.v. capsaicin was reported: (1) an initial drop, followed by (2) a sustained rise in blood pressure, after which (3) another period of hypotension developed (Figure 2) [89]. Bilateral vagotomy abolished the initial, but not the late, depressor response. Furthermore, both depressor responses were absent in rats whose TRPV1-expressing sensory afferents were eliminated by neonatal capsaicin treatment. However, neonatal capsaicin administration paradoxically augmented the second, hypertensive phase of the capsaicin response [89]. Authors concluded that both early and late depressor responses were mediated by capsaicin-sensitive nerves, whereas the rise in blood pressure was caused by a direct vasoconstrictive action of capsaicin on blood vessels [89]. Now, the first and second phases of the capsaicin action are accounted for: the vagally-mediated BJR and the TRPV1-expressing vascular smooth muscle in resistance arteries. One can wonder if the third phase of the capsaicin response was mediated by TRPV1 in vascular endothelium (Figure 2).

## 6. Do Capsaicin-Sensitive Afferents Play a Role in Physiological Blood Pressure Regulation?

As detailed above, in experimental animals, capsaicin injected i.v. evokes a complex and species-dependent action on blood pressure. The first phase of this capsaicin action, triggering the BJR, is clearly relevant to human pathology. Less clear is the role of TRPV1-expressing pathways in physiological blood pressure regulation. This can be studied by the pharmacological blockade of TRPV1, or in animals with genetic disruption or chemical (capsaicin or resiniferatoxin) TRPV1 inactivation. Prior to the molecular cloning of TRPV1, studies relied on neonatal or adult capsaicin “desensitization”. Of note, neonatal and adult desensitization procedures have some important differences. First, neonatal treatment eliminates capsaicin-sensitive neurons (permanent action) [90,91,92,93], whereas adult desensitization is (at least to some degree) reversible [94,95]. Second, neonatally treated animals, similar to *Trpv1*-null animals, can develop compensatory mechanisms [77].

Rats that underwent chemical ablation by capsaicin of TRPV1-positive afferents after birth (50 mg/kg capsaicin s.c.) had normal resting blood pressure values and heart rate [96,97]. Similarly normal resting systemic blood pressure was detected in rats following adult capsaicin desensitization (4, 8 and 16 mg per animal) [24]. Furthermore, small molecule TRPV1 antagonists had no measurable effect on the resting blood pressure, neither in experimental animals nor in human subjects [98,99]. In clinical studies, these antagonists caused hyperthermic reactions and minor burn injuries (due to impaired noxious heat sensation) as adverse effects but did not raise safety concerns about blood pressure regulation [99]. Taken together, these observations imply that TRPV1-sensitive pathways are not involved in maintaining resting blood pressure.

The arterial baroreceptor is critically involved in blood pressure regulation. This pathway has a TRPV1-positive component that can be eliminated by resiniferatoxin (200 μg/kg i.p.) administration [100]. In the resiniferatoxin-treated mice, blunted depressor reflex was noted in response to i.v. phenylephrine infusion [100]. This observation implies a central role for TRPV1-expressing sensory nerves in detecting a rise in blood pressure and triggering countermeasures to maintain homeostasis [100].

The molecular mechanism by which TRPV1 responds to changes in blood pressure remains to be determined. The ability of TRPV1 to directly sense changes in pressure is controversial [101]. For example, pro-opiomelanocortin neurons respond to changes in osmotic pressure in *Trpv1* wild-type, but not in knock-out, mice [102]. Pressure is, however, not a known activator of TRPV1 in sensory neurons [102,103]. TRPV1-positive afferents, however, process and transmit mechanosensitive information, at least under pathologic (e.g., inflammatory) conditions [104,105,106].

Unfortunately, with regard to the exercise pressor reflex (elicited by ventral root stimulation in de-cerebrate mice), conflicting results were reported: one group described a greatly attenuated response in TRPV1-null animals [78], whereas a second study found no effect by pharmacological receptor blockade [107]. Therefore, the role of TRPV1 in sensing blood pressure rise during physical activity remains to be clarified.

## 7. Capsaicin Effects on Blood Pressure in Experimental Models of Hypertension

In the sensory (dorsal root) ganglia of spontaneously hypertensive (SHR) rats, TRPV1 receptor protein, but not mRNA, levels were higher than in normotensive animals, suggesting abnormal posttranscriptional modification in hypertension [108]. In SHR rats, sensory deafferentation by adult s.c. capsaicin administration reduced the mean arterial pressure compared to solvent controls [109]. This was attributed to the depletion by capsaicin of substance P in peripheral sympathetic ganglia [110]. Another major player may be CGRP. CGRP released from capsaicin-sensitive afferents is known to reduce blood pressure [111], and an age-related decrease in CGRP synthesis was implicated in the maintenance of hypertension in SHR rats [112]. (It should be noted here that in clinical studies sustained CGRP inhibition had no effect of the blood pressure of study participants [113].)

In the mouse kidney, approximately half of the glomeruli have closely apposed TRPV1- and/or CGRP-positive nerve endings [114]. These periglomerular sensory afferents are believed to play an important role in the neural control of glomerular filtration of blood pressure regulation.

In the one kidney wrap model of renovascular hypertension, the right kidney is removed and the blood supply of the left kidney is compromised by placing a ligature. In this model, intrathecal capsaicin administration depleted spinal substance P and CGRP and exacerbated the rise in blood pressure [115]. By contrast, intrathecal capsaicin had no effect on the development of desoxycorticosterone (DOCA)-induced hypertension [115]. A second study, however, reported that DOCA-induced hypertension developed earlier and was of greater magnitude in rats with capsaicin treatment as neonates [116].

To explain these seemingly contradictory findings, one should keep in mind that intrathecal capsaicin depleted substance P and CGRP from the dorsal horn of the spinal cord only [115], whereas neonatal capsaicin also eliminated the substance P- and CGRP-positive afferents [116]. Adding to the confusion, mean arterial pressure followed by telemetry was similar in the DOCA-induced hypertension model between the TRPV1-null and wild-type mice [117]. However, the end-organ damage induced by hypertension (including glomerulosclerosis, tubulointerstitial injury and chronic inflammation) was more severe in the TRPV1 knock-out group [117,118]. Based on these observations one may conclude that although TRPV1-sensitive renal afferents may, or may not, play a role in DOCA-induced hypertension, they are likely to constitute a protective mechanism against kidney damage.

Chemical ablation of TRPV1-expressing neurons by neonatal capsaicin (50 mg/kg s.c.) administration rendered the animals salt-sensitive: the treated animals became hypertensive on high salt but not on normal salt diet [119]. The natriuretic response was impaired in the capsaicin-treated animals, as evidenced by the abnormally low Na^+^ concentration in the urine [120]. Based on these findings, authors concluded that the intact sensory afferents were essential for removing the Na^+^ load; in other words, impaired sensory afferents may promote hypertension if a sodium-rich diet is consumed.

In Dahl salt-sensitive (DS) rats kept on a high-salt diet for 3 weeks, reduced TRPV1- and CGRP-like immunoreactivity was detected in the mesenteric resistance arteries compared to DS rats on a low-salt diet [121]. In these animals, the capsaicin-induced urine flow rate, glomerular filtration rate and perfusion pressure were all diminished compared to Dahl salt-resistant rats or DS rats on a low-salt diet [122].

In men, obesity is often associated with hypertension. To study the role of TRPV1 in obesity-associated hypertension, wild-type and TRPV1-null mice were kept on a high-fat diet for 3 to 15 weeks: both groups became obese on the high-fat diet, but only wild-type animals developed hypertension and cardiac hypertrophy [123]. According to this study, the TRPV1-expressing innervation may play a central role in the development of obesity-induced hypertension. If so, TRPV1 receptor antagonists may represent a new class of antihypertensive drugs.

Unexpectedly, functional TRPV1 was detected in both murine [124] and human adipocytes [125]. Mice lacking UCP1 (uncoupling protein-1) represent a genetic model of spontaneous obesity: these animals become obese and hypertensive when they grow old [126,127]. In the TRPV1/UCP1 double-knock-out mice, both obesity and hypertension developed at a younger age and were more severe than in the UCP1-null animals [126], implying a protective role of TRPV1 against obesity-induced hypertension. This study, if the results hold true in humans, raises concerns about the safety of TRPV1 antagonists.

## 8. Dietary Capsaicin and Blood Pressure in Animal Studies

The effect of dietary capsaicin on experimental hypertension was studied in wild-type and TRPV1-null (*Trpv1*^−/−^) male C57BL/6J mice [128]. Hypertension was induced by a high-salt diet (8% NaCl) administered with or without capsaicin (0.01%) for 12 weeks. Blood pressure was determined by telemetry. Dietary capsaicin prevented the salt-induced nocturnal rise in blood pressure in the wild-type but not the TRPV1-null animals; in fact, no difference in blood pressure was noted between the high-salt diet supplemented with capsaicin and the control (normal salt diet, 0.4% NaCl) groups [128]. In hypertensive mice, reduced NO levels were determined in the mesenteric arteries. Dietary capsaicin normalized the endothelial NO production [128]. The authors concluded that dietary capsaicin may prevent the endothelial dysfunction caused by salt-induced hypertension. Furthermore, they argued that dietary capsaicin may represent a promising nutraceutical in patients with salt-sensitive hypertension [128]. The same group described a second molecular mechanism by which dietary capsaicin may exert an antihypertensive action in mice fed a high-salt diet: capsaicin inhibited Na^+^ reabsorption in the kidney by blocking the activity of the epithelial Na^+^-channel [129]. This is in keeping with the abnormally low Na^+^ concentration in the urine of animals following neonatal capsaicin desensitization [118].

In the 2-kidney, 1-clip model of renovascular hypertension, the administration of dietary capsaicin (0.006%) for 6 weeks reduced the systolic blood pressure compared to a bland diet [130]. The capsaicin effect on blood pressure rise was abolished by co-administration of the endothelial nitric oxide synthase (eNOS) inhibitor, L-NAME (N-Nitro-L-arginine methyl ester hydrochloride) [130], implicating endothelial NO production in the beneficial capsaicin effect.

## 9. Human Observations

In his authoritative treatise on hot peppers, the German physician and botanist Karl Anton Fingerhuth (1798–1876) provided the first detailed description of capsaicin actions in humans [131]. He described the sensory irritation and gastrointestinal disturbances but made no mention of any change in heart rate or blood pressure. In 1877, Hőgyes tested “capsicol” (a capsaicin-containing oily pepper extract) on himself and observed no change in his heart rate [21], despite the flatulence and diarrhea already documented by Fingerhuth. In 1986, capsaicin (0.5 μg/kg) was injected into the superior vena cava or an arm vein of three volunteers: all study participants reported a “burning” chest feeling, but no change in heart rate, blood pressure or tidal volume was detected [132]. However, in one volunteer, a higher capsaicin dose (4 μg/kg) provoked paroxysmal coughing [132]. According to this study, in conscious healthy men i.v. capsaicin does not trigger the BJR, although it stimulates the nociceptive pulmonary afferents. This is surprising since i.v. capsaicin evoked the full BJR in all species tested [21,23,24,40,41,42] with the sole exception of the guinea-pig [43], and even in guinea-pigs capsaicin induced a brief drop in blood pressure [43].

In healthy adult volunteers (fifteen men, aged 21–33), intradermal capsaicin (10 μg) injected into the volar aspect of the forearm at a depth of 5 mm increased both systolic and diastolic blood pressures compared to topical administration [133].

Qutenza is a high-concentration (8%) topical capsaicin patch used to relieve chronic neuropathic pain. In patients with neurofibromatosis-1 (NF1), Qutenza was well-tolerated with no detectable effect on blood pressure [134]. However, in other studies a few patients experienced a transient increase in blood pressure [135]. Conversely, the topical capsaicin patch was found to improve coronary flow in patients with stable angina [136]. This effect was attributed to improved endothelial NO production [137].

Intravesical capsaicin is in clinical use to decrease the number of incontinent episodes in patients with detrusor hyperreflexia due to spinal cord injury [138]. In this patient population, intravesical capsaicin (2 mM) applied for 30 min resulted in a rise in blood pressure with a peak observed 10 min after instillation; then the blood pressure returned to normal within 40 min [139].

In a randomized, controlled clinical trial with the “modality-selective” TRPV1 antagonist, NEO06860, in patients with osteoarthritis knee pain, elevated blood pressure was reported as a possible adverse effect in 3 out of the 57 study participant (5% of the patients) [140].

However, in healthy volunteers, NEO06860 had no measurable effect on resting blood pressure [141].

The China Health and Nutrition Survey involving 9273 participants reported somewhat lower resting blood pressure (and a lower risk for developing hypertension) in women who consumed hot spicy food frequently, but not in men [20]. The mean systemic blood pressure in women who ate spicy food at least five times a week was 120.8 mmHg; this should be compared to the value of 126.4 mmHg in women who did not consume spicy meals at all. The odds ratio for developing hypertension was 0.546 in the chili-eater group compared to non-eaters [20].

Last, ingesting extreme amounts of hot pepper was reported to provoke a fatal coronary vasospasm in a young individual [142].

## 10. Discussion

In animal experiments, i.v. capsaicin evokes a characteristic, biphasic response: (1) a brief hypotensive response (accompanied by bradycardia and slow, shallow breathing), followed by (2) a sustained rise in blood pressure [24,25,41,42,43]. In some studies, a period of late hypotension was also noted [89]. The initial depressor response is attributed to triggering the Bezold–Jarisch reflex, also known as the pulmonary chemoreflex [25,40], whereas the pressor response most likely represents a direct capsaicin action on TRPV1 expressed in vascular smooth muscle of the resistance arteries (Figure 2) [71,72,73,74,75,76,77]. Less clear is the mechanism underlying the late vasodilator phase. It may reflect rebound due to the removal of the myogenic tone, but CGRP released from sensory afferents and/or increased NO production by capsaicin in the endothelium might also play a role (Figure 2) [87,88].

The gamut of experimental evidence indicates that the capsaicin receptor TRPV1 is not involved in physiological blood pressure regulation in experimental animals. Indeed, the resting blood pressure of rodents was normal following neonatal [96,97] or adult [24] capsaicin desensitization and/or inactivation of the *Trpv1* gene by genetic manipulation [128].

Much less known is the effect of capsaicin on blood pressure regulation in humans. In three volunteers, i.v. capsaicin evoked a “burning” sensation in the chest, but no change in heart rate, blood pressure or tidal volume [132]. If this small sample is representative, capsaicin does not evoke the Bezold–Jarisch reflex in men. However, capsaicin injected deep into the dermis [133], or applied via catheter into the urinary bladder [139], did evoke a sustained rise in blood pressure, indicating that the second phase of the capsaicin response (the direct vasoconstrictor action) is also functional in men. If capsaicin (a TRPV1 agonist) can increase human blood pressure, one may argue that a TRPV1 antagonist could have the opposite effect. This prediction could easily be tested in hypertensive patients. (Parenthetically, in rats the first generation TRPV1 blocker, BCTC (N-(4-tertiarybutylphenyl)-4-(3-chloropyridin-2-yl)tetrahydropyrazine-1(2H)-carbox-amide), evoked a drop in blood pressure [77]).

Epidemiological studies have established a link between eating chili pepper regularly and having a normal blood pressure [7,8,9,20]. In keeping with this, dietary capsaicin was found to reduce blood pressure in various experimental models of hypertension [128,129,130]. Alternatively, this could represent an indirect capsaicin effect on blood pressure regulation. Indeed, enjoyment of spicy food was shown to enhance salty taste in men, and thereby reduce salt intake [143]. Since a salt-rich diet is a known risk factor for elevated blood pressure, it is easy to visualize how capsaicin can help maintain a healthy blood pressure by reducing salt intake.

In summary, TRPV1 seems to play an important but as yet poorly understood role in blood pressure regulation. Animal experiments suggest a role for TRPV1 in hypertension, but human validation is still absent. Clinical studies with the already available potent and selective small molecule TRPV1 antagonists may provide the answer to the opening question of this review: can TRPV1 represent a novel therapeutic target in hypertension?

## Figures and Tables

**Figure 1 ijms-24-08769-f001:**
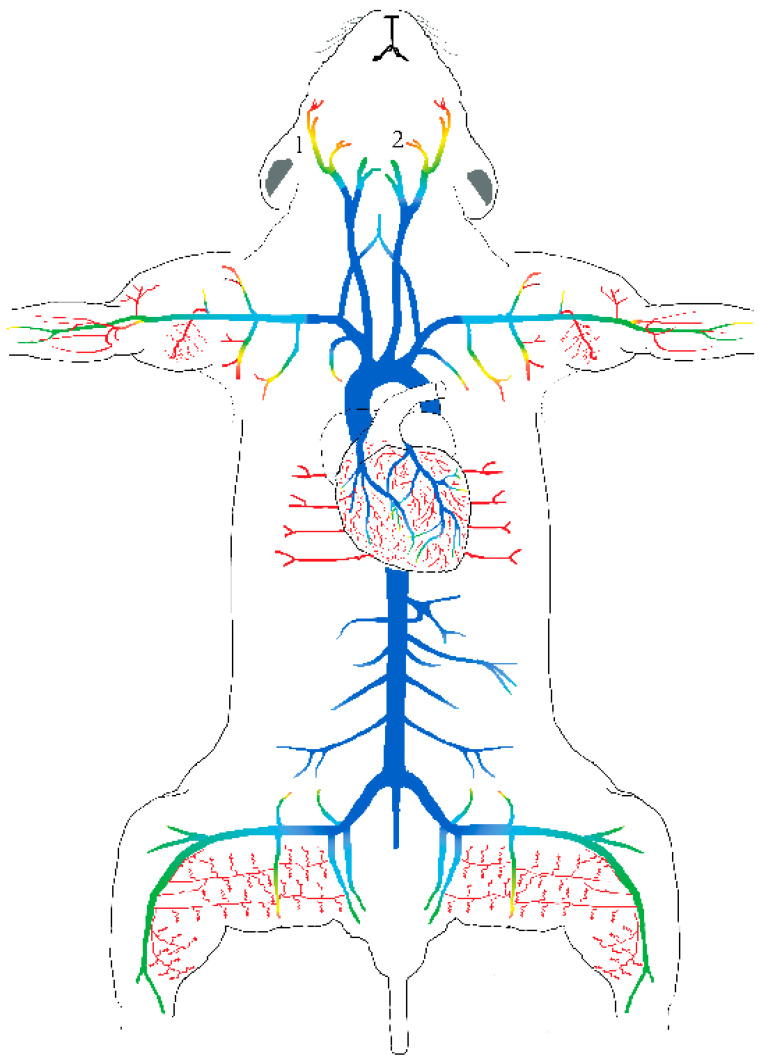
Arterial heat-map of functional TRPV1 expression, as revealed by the TRPV1^PLAPnLacZ^ reporter mouse. 1. Superficial temporal artery; 2. facial artery. Reprinted/adapted with permission from [76]. Copyright owner: John Wiley and Sons.

**Figure 2 ijms-24-08769-f002:**
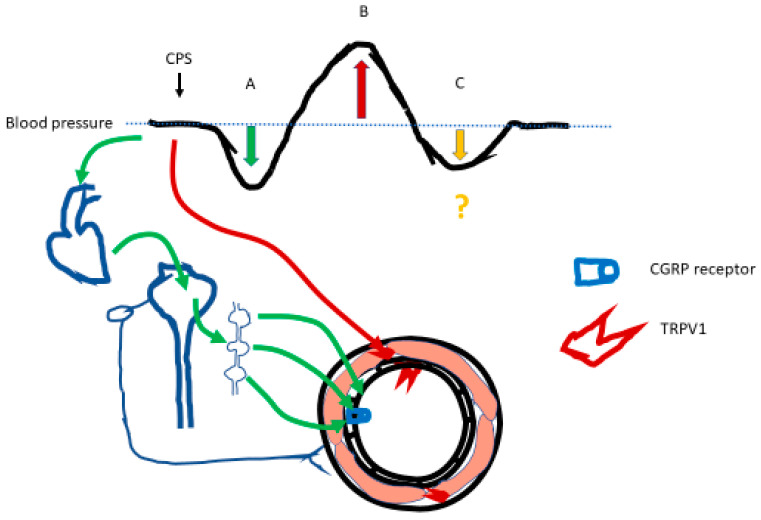
Schematic representation of the tri-phasic capsaicin effect on blood pressure in experimental animals, along with the presumptive underlying molecular mechanisms of action (Figure courtesy of Attila Tóth). (A) When injected intravenously, capsaicin (CPS) evokes the Bezold–Jarisch reflex (green line); this produces a rapidly developing, but transient drop in blood pressure. The Bezold–Jarisch reflex is now thought to originate from cardiac inhibitory mechanoreceptors located in the left ventricle of the heart. The efferent reflex pathway, that runs in the vagal nerve, increases the parasympathetic tone. (B) Circulating capsaicin binds to TRPV1 receptors expressed on vascular smooth muscle (red line). This evokes a direct vasoconstrictive effect by allowing Ca^2+^ influx, increasing the blood pressure. (C) The molecular mechanism underlying the late drop in blood pressure (yellow arrow) is yet to be elucidated. To some extent, this may reflect a rebound effect that follows the disappearance of the myogenic tone. Alternative mechanisms include the CGRP effect (released from sensory afferents) on the vasculature and/or direct capsaicin action on endothelial cells.

## Data Availability

Not applicable.

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
