# Peer review of "The Vanilloid (Capsaicin) Receptor TRPV1 in Blood Pressure Regulation: A Novel Therapeutic Target in Hypertension?"

_ijms, 2023, doi:10.3390/ijms24108769_

Round 1

Reviewer 1 Report

In general, this is a very nice, detailed and historical account of TRPV1 pharmacology and the effects on blood pressure.  I have a couple of suggestions. First, I think the author should mention the limitations with TRPV1 antibodies and how non-specificity may have contributed to uncertainty about the precise TRPV1 expression pattern. In particular, the TRPV1 reporter mice do not show strong evidence for vascular endothelial TRPV1 expression. Second, with respect to the third phase of the capsaicin evoked BP response, the author raises the possibility for a neurogenic mechanism (CGRP). I think another possibility is the transient removal of myogenic tone following the capsaicin induced shortening of the vascular smooth muscle. Indeed, this is a possible explanation for the rebound dilation shown in Fig. 3. 

Minor:

1.     Figure 1 in the text (P.8) is erroneously referenced to #89; it should be #76.

2.     P 12, “BCTC, evoked a drop in blood pressure” should refer to Figure 4 not Figure 5.

Author Response

Dear Referee:

Thank you for your comments. I fully agree, differences in the reported TRPV1 protein expression to large degree may reflect differences in the sensitivity and specificity of the anti-TRPV1 antibodies used by the various research groups. Now this is mentioned in the text (statement highlighted in yellow). Just for your information, we have tested and compared 8 commercially available anti-TRPV1 antbodies and detected problems with 6! These are yet unpublished results, therefore I did not want to go into details in this MS.

I also agree with your alternative explanation of the second phase of vasodilation: rebound due to removal of the myogenic tone. Now this is mentioned in the text (highlighted in yellow). 

The two minor problems were corrected. 

Reviewer 2 Report

The author, Dr. Szallasi is a leading researcher of TRP channels, especially TRPV1. This review includes cutting-edge knowledge from basic to recent findings and will attract a large number of readers of Int J Mol Sci. However, there are several concerns to discuss before accepting this.

1. Figures 1, 2, 3, and 4: The author should supply high-resolution data.

2. Figures 1 to 4 are important figures but lack impact. The author must add one or more impressive schematic diagrams to increase citations by the readers.

3. The following abbreviations are only used once: POMC (Page 8), 1K-WRAP (Page 9), 2K1C (Page 10), and NF1 (Page 11). The author should remove them.

4. Please supply the chemical name of BCTC.

5. Which is correct, NE06860 or NEO06860?

Author Response

Dear Reviewer:

Thank your for your comments. I fully agree with the need for publication quality, high resolution figures: these will be submitted separately. For some reason, the figures lost most of their resolution when inserted into the MS Word file. 

Sadly, my computer graphic skills are non-existent, and MDPI does not provide a professional computer graphic service either. Therefore, I cannot generate the "high impact" figure that you suggested. Sorry.

The superfluous abbreviations were removed.

The chemical name of BCTC was added. 

It is correctly NEO06860.

Reviewer 3 Report

The author presents a narrative review manuscript of the literature that attempts to continue a very classic research theme. The author makes a very simple and little novel exposition. The author limits himself to making an exposition that is more informative than scientific. The figures used are not adequately justified, and they are not their own. The references provided are more than 70% obsolete. This manuscript is not of adequate quality to be published in this Journal. The manuscript is not adequately written in the scientific quality that a narrative review contributed by an expert in the field should have.

Author Response

Reviews represent a very subjective form of art - some like the outcome, whereas others (like you) apparently do not. Referee 1 described this MS as "very nice, detailed" and referee 2 characterized it as "cutting edge knowledge from basic to recent." 

Old does not make anything "obsolete." According to the Webster dictionary, "obsolete" implies something useless. I do not consider any of the studies cited in my review "useless." Capsaicin research has a rich and exciting history, going back to the seminal observations of Endre Hogyes more than a hundred years ago. 

Last but not least: I had the MS evaluated by a native English speaking colleague. He found the MS ease to read. He suggested a few minor stylistic changes that I implemented. 

Round 2

Reviewer 2 Report

Because the author could not revise adequately, the reviewer could not satisfy the author's comments.

Author Response

Dear Referee:

As per your request, the new version of the MS (2nd revision) now includes a figure (Figure 2) with the schematic representation of the capsaicin effect on blood pressure (3 phases), along with their presumptive underlying molecular mechanisms of action. 

The previous figures were removed from the MS, with the sole exemption of the heat-map of the arterial TRPV1 expression (now Figure 1). 

Your other comments were already addressed in the 1st revision.

I hope that with these changes now you find the MS acceptable for publication.

Respectfully,

Arpad Szallasi

Reviewer 3 Report

The authors have not made any changes from the previous version. The authors present a review of little or no quality.

Author Response

Dear Referee:

All the points raised by the other two referees have been addressed in the 2nd version of the revised MS.

To address your comment on English, the MS was read and corrected by a native English speaker.

Since you did not specify the references that you considered obsolete, I was unable to consider those for elimination.

Respectfully,

Arpad Szallasi